# Effects of Light Shading, Fertilization, and Cultivar Type on the Stable Isotope Distribution of Hybrid Rice

**DOI:** 10.3390/foods12091832

**Published:** 2023-04-28

**Authors:** Syed Abdul Wadood, Yunzhu Jiang, Jing Nie, Chunlin Li, Karyne M. Rogers, Hongyan Liu, Yongzhi Zhang, Weixing Zhang, Yuwei Yuan

**Affiliations:** 1State Key Laboratory for Managing Biotic and Chemical Threats to the Quality and Safety of Agro-Products, Hangzhou 310021, China; abwadood.fn@uhe.edu.pk (S.A.W.);; 2China National Rice Research Institute, Hangzhou 310006, China; 3Institute of Agro-Product Safety and Nutrition, Zhejiang Academy of Agricultural Sciences, Key Laboratory of Information Traceability for Agricultural Products, Ministry of Agriculture and Rural Affairs of China, Hangzhou 310021, China; 4Department of Food Science, University of Home Economics Lahore, Lahore 54700, Pakistan; 5National Isotope Centre, GNS Science, 30 Gracefield Road, Lower Hutt 5040, New Zealand; 6Research Center for Plants and Human Health, Institute of Urban Agriculture, Chinese Academy of Agricultural Sciences, Chengdu 610213, China

**Keywords:** light intensity, fertilizer treatment, stable isotopes, hybrid rice, cultivar, fractionation mechanism

## Abstract

The effect of fertilizer supply and light intensity on the distribution of elemental contents (%C and %N) and light stable isotopes (C, N, H, and O) in different rice fractions (rice husk, brown rice, and polished rice) of two hybrid rice cultivars (maintainer lines You-1B and Zhong-9B) were investigated. Significant variations were observed for *δ*^13^C (−31.3 to −28.3‰), *δ*^15^N (2.4 to 2.7‰), *δ*^2^H (−125.7 to −84.7‰), and *δ*^18^O (15.1‰ to 23.7‰) values in different rice fractions among different cultivars. Fertilizer treatments showed a strong association with %N, *δ*^15^N, *δ*^2^H, and *δ*^18^O values while it did not impart any significant variation for the %C and *δ*^13^C values. Light intensity levels also showed a significant influence on the isotopic values of different rice fractions. The *δ*^13^C values showed a positive correlation with irradiance. The *δ*^2^H and *δ*^15^N values decreased with an increase in the irradiance. The light intensity levels did not show any significant change for *δ*^18^O values in rice fractions. Multivariate ANOVA showed a significant interaction effect of different factors (light intensity, fertilizer concentration, and rice variety) on the isotopic composition of rice fractions. It is concluded that all environmental and cultivation factors mentioned above significantly influenced the isotopic values and should be considered when addressing the authenticity and origin of rice. Furthermore, care should be taken when selecting rice fractions for traceability and authenticity studies since isotopic signatures vary considerably among different rice fractions.

## 1. Introduction

Rice (*Oryza sativa* L.) is a major cereal crop consumed by half of the global population and widely planted in Asia, Africa, and parts of America. In 2020, the total area under rice cultivation exceeded 195 million hectares. The quality characteristics of rice are mainly associated with its growing conditions, and recently the traceability of rice back to its growing origins has gained increasing interest from consumers, producers, and related industries since it is vulnerable to economic fraud [1]. Many methods have been developed to address the authenticity of rice, including multi-element, spectroscopic, omic, and DNA-based analysis [2,3,4,5]. Most recently, stable isotope analysis (SIA) has been widely employed to authenticate organic rice, determine its geographic origin, and identify rice cultivars [6,7]. However, there may be unknown stable isotope effects on rice, caused by cultivar type, light intensity, environmental factors, and fertilizer treatments which may reduce the accurate determination of its geographical origin, especially when it is procured from nearby or adjoining localities. Therefore, identifying the range of stable isotope compositions (*δ*^13^C, *δ*^15^N, *δ*^2^H, and *δ*^18^O) in different rice fractions according to the differences in light intensity level, fertilizer type and concentration, and cultivar/variety would improve the validity of traceability methods for rice and its products.

Different factors such as plant physiology, photosynthetic processes, climatic factors (temperature, sunshine, humidity, precipitation), and cultivation practices have been shown to induce stable isotope fractionation in plants [8]. Generally, the *δ*^13^C values of plants reflect plant photosynthetic processes and water use efficiency. Carbon found in mature rice grains originates from the assimilation of CO_2_ during the grain filling period [9]. Solar irradiance is the main factor that affects the net CO_2_ assimilation rate and high temperature is also associated with stomatal closure and a reduction in CO_2_ assimilation. Conversely, if the temperature is below the optimum range, the net CO_2_ assimilation rate will become light-limited and lead to reduced photosynthetic productivity [10]. In addition, plant photosynthesis is also affected internally by photozymes, hydrolase C3 reductase, and CO_2_ fixation enzymes [11]. This internal physiological response is referred as the physiological index which mainly reflects stomatal conductance, transpiration rates, net photosynthetic rate, and intercellular CO_2_ concentration. Plant *δ*^2^H and *δ*^18^O values reflect physical factors such as rainfall and evapotranspiration, and are also associated with plant physiological parameters such as transpiration and stomatal conductance [12]. Nitrogen isotopes are mostly associated with farming practices, crop types, and soil characteristics and also reflect significant correlations between plant growth, photosynthetic capacity, and respiration rate [8,12].

Many studies have reported the limitation of stable isotopes in addressing the authenticity of agro-products when the samples are procured from close geographical locations or from a region where the same agricultural practices (fertilizer, etc.) are adopted [13]. Therefore, it is very important to understand the effects of different factors (light intensity, fertilizer type and concentration, and cultivar type) on the composition of rice stable isotopes. To explore these effects, we conducted two field experiments with a split plot design using two commonly used Chinese hybrid rice cultivars (maintainer lines) Zhong-9B and You-1B. The objective of this study was to investigate the variability of %C, %N, *δ*^13^C, *δ*^15^N, *δ*^2^H, and *δ*^18^O values in different rice fractions (husk, polished rice, and brown rice) in response to different fertilizer regimes and light intensity levels, and to identify how light shading and fertilizer application control isotopic ratios in rice. The results from this study will contribute more insight into the localized climatic, environmental, and farming effects on the isotopic composition of rice and will allow us to better predict rice origin and authenticity using stable isotope-based traceability models.

## 2. Material and Methods

### 2.1. Field Experiment

Field trials were carried out in 2018 at an experimental field at the China National Rice Research Institute (CNRRI) in Fuyang, Zhejiang province. Two commonly used hybrid rice maintainer lines (Zhong9B and You1B, which are the most popular cultivars in China provided by CNRRI), were studied. In total, three nitrogen treatments (N0, N6, and N12) with three light intensities (shading; 50% and 75%, and non-shading; ambient light) were arranged in a split plot design. The size of each plot was 10 m^2^ with 3 replicates for each treatment. Nitrogenous fertilizer was applied as 0 kg/ha (N0), 90 kg/ha (N6), and 180 kg/ha (N12), respectively, at different growing stages including basal planting, tillering, and heading stages which accounted for 50%, 30%, and 20% of the application, respectively. Phosphate fertilizers were used only as basal fertilizer, whereas potash was used as both basal and tillering fertilizers and the fertilizer proportion was N:P:K = 1:0.5:0.5, respectively. 

In the case of N0, urea was not applied during the entire production period. In total, 450 kg/ha of super phosphate was applied as basal fertilizer and 150 kg/ha of potash as basal (75 kg/ha) and tillering stage (75 kg/ha), respectively. For N6 treatment, 195 kg/ha of urea was applied, including 97.5kg/ha as basal fertilizer, 58.5 kg/ha as tillering fertilizer (7 to 9 days after transplanting), and 39 kg/ha as heading fertilizer (30 days after transplanting), respectively. In the case of N12 treatment, a total of 390 kg/ha of urea was applied, of which 195 kg/ha was basal fertilizer, 117 kg/ha was tillering fertilizer, and 78 kg/ha was applied as heading fertilizer. The application rate of phosphate and potash fertilizers for both N6 and N12 treatments were the same as N0. 

For the light intensity investigation, natural sunlight treatment, LS-0 (0% shading, no shading), and shaded treatments, LS-50 (50% shading) and LS-75 (75% shading) were applied. In the shaded treatments, plants were shaded with a shading screen/net. For LS-50, one layer was used and for LS-75 two layers of shading screen were applied over the top of the rice plants. 

### 2.2. Elemental Content and Isotope Ratio Measurements

Rice grains (2 kg) from each plot were harvested at maturity and subsequently threshed by a hulling machine equipped with a rice polisher (PY-200, Hubei Pinyang Technology Co., Ltd., Xiaogan, China) to obtain different fractions including brown rice (BR), polished rice (PR), and rice husk (RH). All fractions were air-dried, then ground into a fine powder, and finally dried at 50 ± 2 °C for 24 h. The samples were stored in desiccators until further analysis. For the determination of elemental contents (% C, % N) and isotopes (*δ*^13^C and *δ*^15^N), dried powdered samples were weighed (4.5 to 5.5 mg) and packed into tin capsules (3 mm × 5 mm). The samples were combusted in an elemental analyzer (Vario Pyro Cube, Elementar, Hanau, Germany) and the combustion gases were analyzed using an isotope ratio mass spectrometer (IRMS) (IsoPrime100, Isoprime Ltd., Manchester, UK). Sample combustion was carried out in a combustion furnace at 1150 °C and reduction in the N_2_O_x_ gases to N_2_ over copper wire occurred at 850 °C. An inert gas (He) with a flow rate of 230 mL/min was passed through a CentrION prior to mass spectrometry. Acetanilide (Puriss. p.a., Sigma-Aldrich) was used to calibrate elemental % C and % N. For *δ*^13^C and *δ*^15^N analysis, multipoint calibration was applied using reference standard materials including B2155 (protein, *δ*^13^C = −27.0‰, *δ*^15^N = +6.0‰), IAEA-CH-6 (sucrose, *δ*^13^C = −10.4‰), USGS40 (L-glutamic acid, *δ*^13^C = −26.4‰, *δ*^15^N = −4.5‰), USGS64 (glycine, *δ*^13^C = −40.8‰, *δ*^15^N = +1.8‰), and IAEA-N-2 (ammonium sulfate, *δ*^15^N = +20.3‰). The *δ*^13^C and *δ*^15^N values were measured relative to V-PDB and AIR, respectively. 

For *δ*^2^H and *δ*
^18^O isotopes, around 1.0 mg powdered sample of each fraction was weighed into silver capsules (6 mm × 4 mm) and analyzed using EA (Vario Pyro Cube, Elementar, Hanau, Germany) IRMS (IsoPrime100, Isoprime Ltd., Manchester, England). Reference materials USGS54 (Canadian lodgepole pine, *δ*^2^H = −150.4‰, *δ*^18^O = +17.8‰) and USGS55 (Mexican ziricote, *δ*^2^H = −28.2‰, *δ*^18^O = +19.1‰) were used to calibrate the *δ*^2^H and *δ*^18^O measurements. Samples and reference materials were freeze-dried at −60 °C for three days to remove all adsorbed water and subsequently equilibrated for five days in the laboratory and exposed to local atmospheric conditions prior to H and O analysis. Pyrolysis was performed at 1450 °C to convert organic H and O to gaseous H_2_ and CO, respectively, and finally the analytes were transferred into the IRMS for isotope determination. The *δ*^2^H and *δ*^18^O values were measured relative to Vienna Standard Mean Ocean Water (V-SMOW). All the samples were analyzed in triplicate. Reference materials were sourced from the International Atomic Energy Agency (IAEA, Vienna, Austria) and the United States Geological Survey (USGS, Reston, Virginia, United States). B2155 was supplied by Elemental Microanalysis (Okehampton, United Kingdom). The analytical precision for *δ*^13^C, *δ*^15^N, *δ*^2^H, and *δ*^18^O was less than ±0.1‰, ±0.2‰, ±2‰, and 0.5‰, respectively. The delta values (*δ*) were calculated as follows:(1)$$\delta E=\left(R_{\mathrm{sample}}\;/{\;R}_{\mathrm{standard}}\right)-1$$
where *δ*E represents *δ*^13^C, *δ*^15^ N, *δ*^2^H, and *δ*^18^O whereas R_sample_ and R_standard_ represent the ^13^C/^12^C, ^15^N/^14^N, ^2^H/^1^H, or ^18^O/^16^O ratios in samples and reference materials, respectively. 

### 2.3. Data Analysis

The effect of light intensity, fertilizer type and concentration, cultivars, and their interaction were studied using %C, %N, *δ*^13^C, *δ*^15^ N, *δ*^2^H, and *δ*^18^O of different rice fractions with multivariate analysis of variance. Light shading level, fertilizer type and concentration, and cultivars were considered as fixed variables. Differences among the treatments were evaluated using Duncan’s test at a significance level of 0.05. All analyses were performed using R software (version 3.0.3).

## 3. Results and Discussion

### 3.1. Multivariate ANOVA for Elemental Content and Isotope Ratios

Multivariate analysis of variance was applied to evaluate the effect of different factors such as variety (vty), shading level, and fertilizer (nitrogen) concentration on the carbon (%C), nitrogen (%N), *δ*^13^C, *δ*^15^N, *δ*^2^H, and *δ*^18^O values of different rice fractions, including polished rice (PR), brown rice (BR), and rice husk (RH). The results are summarized in Table 1 and shown in Figure 1. The results showed that the carbon content (%C) in different rice fractions was not affected by vty, light shading, fertilizer, or their interactions. For %N, light shading showed a significant influence on the total nitrogen content of PR and BR; however, no significant difference was observed in RH under different light shading levels. Moreover, an interaction (vty × light shading) effect was also observed for PR (Figure 1a). In the case of *δ*^13^C, light shading and variety significantly contributed to all rice fractions. The interaction (vty × fertilizer)/(vty × light shading) effect also showed significant influence for PR and (vty × light shading) for RH. No interaction effect was observed for BR (Figure 1b–d). 

The *δ*^15^N values in RH, PR, and BR were significantly affected by fertilizer concentration, shading level, and interaction (vty × shading level) effects (Figure 1e–g). Significant interaction effects (vty × fertilizer) on *δ*^15^N were also observed for PR (Figure 1h) and (fertilizer × light) for RH (Figure 1i). The fertilizer concentration and shading levels were the major factors that contributed significant variation among all rice fractions. In the case of *δ*^2^H, the fertilizer concentration and shading level showed a significant difference for PR and BR, whereas no significant difference was observed for RH. The interaction (vty × fertilizer)/(vty × light shading) effect was only observed for PR (Figure 1j,k). Only cultivar imparted a significant variation for the *δ*^2^H values in RH. In the case of *δ*^18^O, different trends were observed. Almost all factors contributed to a significant variation for RH but no significant differences were observed for BR and PR. The interaction (vty × fertilizer)/(vty × light shading) effects for *δ*^18^O are shown in Figure 1l,m, respectively.

### 3.2. Elemental and Isotope Differences between the Rice Varieties

Differences in the %C, %N, *δ*^13^C, *δ*^15^N, *δ*^2^H, and *δ*^18^O values among the different rice cultivar fractions under different fertilizer concentrations and light intensities were determined. The results showed significant differences for all the analyzed parameters except for %C. Multiple comparisons were made between %N, *δ*^13^C, *δ*^15^N, *δ*^2^H, and *δ*^18^O values (Figure 2). The %N in RH showed a strong significant difference between the two rice varieties. The total %N of You-1B rice (1.8 ± 0.2%) was significantly higher than Zhong-9B rice (1.5 ± 0.4%). Similarly, the *δ*^13^C values of You-1B RH (−30.8‰), PR (−28.3‰), and BR (−29.1‰) were significantly higher than Zhong-9B (−31.3‰, −29.2‰, and −29.7‰), respectively. The lower *δ*^13^C values of Zhong-9B rice suggest higher water use efficiency. Different factors are responsible for genetic variations in the *δ*^13^C values, including diffusive conductance, water use efficiency, stomatal activity, etc. [14]. For the *δ*^18^O values, the two varieties also showed significant differences for PR and RH. Higher *δ*^18^O values of You-1B RH (23.7‰) indicate higher transpiration rates. Our results are consistent with previous findings where a similar trend was observed for *δ*^18^O variations among different rice cultivars [15]. Differences in the *δ*^18^O values are mainly due to the vegetative cycle of water [16]. 

A different pattern was observed for the *δ*^2^H values. You-1B BR (−88.6‰) and RH (−125.7‰) were significantly lower than Zhong-9B BR and RH (−84.7‰ and −117.0‰), respectively. This difference suggests that less water fractionation occurred in Zhong-9B rice. A previous study also reported a significant difference in the *δ*^2^H values among different rice cultivars [17]. 

The *δ*^15^N values also followed the same trend where Zhong-9B rice values were higher than You-1B, although the difference in traits between rice varieties was primarily reflected in RH. *δ*^15^N values of PR, BR, and RH in Zhong-9B and You-1B rice was (2.6‰, 2.5‰, 2.7‰) and (2.5‰, 2.4‰, and 2.4‰), respectively. Our results are consistent with previous findings where significant differences in the *δ*^15^N values among different rice cultivars were reported [18].

### 3.3. Effect of Fertilizer Regimes on Elemental and Isotope Values of Rice Fractions

The effect of different fertilizer treatments on the %C, %N, *δ*^13^C, *δ*^15^N, *δ*^2^H, and *δ*^18^O values was measured and the results are shown in Figure 3. %N did not show any significant effect among the rice fractions and the only interaction effect (variety × nitrogen) was detected in PR rice (Table 1). The %N and *δ*^15^N values among different rice fractions were significantly affected by nitrogen fertilizer application levels. The %N and *δ*^15^N values of different rice fractions under different fertilizer concentrations (N0, N6, and N12) followed different patterns of accumulation. Figure 3 shows that the %N of rice grains was higher when subjected to higher nitrogen application levels. The %N for BR (1.8 ± 0.2%) grown under N12 conditions was significantly higher than N6 (1.6 ± 0.26%) and N0 (1.6 ± 0.2%), respectively. However, the *δ*^15^N value for N0 was significantly higher than N6 and N12 indicating that the synthetic urea fertilizer application level was negatively correlated with the *δ*^15^N values in different rice fractions (Figure 3). 

These results appear consistent with previous findings which suggest conventionally fertilized rice has lower *δ*^15^N values compared to control samples (without fertilizer) [19]. The interpretation of *δ*^15^N signatures in plant tissues is generally complex since it can differ from the nutrient source as ^15^N fractionation occurs during plant physiological processes, mainly during nitrogen uptake, nitrate assimilation, and reduction, as well as during remobilization to the rice grain [20,21]. Lower *δ*^15^N values in the fertilized plant fractions during pre-anthesis reflect the direct N availability from the fertilizer applied. Discrimination against ^15^N occurs during the assimilation of inorganic nitrogen within the plant, and the enzymes responsible for this isotope discrimination include nitrate reductase and glutamine synthase [22]. 

Higher N assimilation rates in fertilized plants enhance the uptake of inorganic fertilizer which results in the reduction in ^15^N in fertilized plants compared to non-fertilized plants [22]. In addition, the isotopic discrimination against ^15^N is marginal in non-fertilized rice ears (glumes, awns, grains) as compared to fertilized rice ears, probably because N limitation prevents discrimination and promotes a higher efficiency for N remobilization [23]. The higher amount of N carried over into the grain results in a higher ^15^N discrimination. This fact suggests a decrease in the *δ*^15^N values for the rice grain in fertilized plants compared to non-fertilized plants. Discrimination within the plant is lower in the non-fertilized grains (N0; lacking nitrogen fertilizer) because the ratio between the plant demand vs. available N is high [24].

Nitrogen fertilizer application had a significant effect on the *δ*^2^H values in PR and BR. The highest *δ*^2^H values in PR (−73.2‰) and BR (−83.2‰) were observed for the N12 treatment which indicates that the *δ*^2^H values increased with N fertilizer level. There is limited literature on this topic, which restricts the discussion of this mechanism, although a similar trend was observed for the *δ*^2^H values in a previous study, where it was reported that BS-fertilized (biogas slurry) rice had higher *δ*^2^H values than a control rice (without fertilizer) [19]. The effects of nitrogen application levels on the *δ*^18^O values contrasted with *δ*^2^H values. There was no significant effect in the PR and BR grains, probably due to a weak or no effect on grain morphological parameters, and a lack of N effect on stomatal conductance and transpiration rates [25]. 

A significant difference was observed in the *δ*^18^O values in RH. The mean *δ*^18^O values for N6 (21.3‰) and N12 (20.2‰) treatments were significantly lower than N0 treatments (22.9‰) suggesting that the *δ*^18^O values in RH decreased with increasing nitrogen fertilization. Lower *δ*^18^O values in RH than the rice grain may be associated with progressive enrichment in plant components from the root, to the stem, to the leaf, and finally to the grain, as well as physiological factors such as dehydration and plant tissue degradation that occurs during husk development [26].

### 3.4. Elemental and Isotopic Variations among Different Light Shading Treatments

Different shading treatments including LS-0, LS-50, and LS-75 were applied and the effect was observed on the %N, %C, *δ*^13^C, *δ*^15^N, *δ*^2^H, and *δ*^18^O values of different rice fractions. The experiment was performed using two shading treatments (LS-0 and LS-50) for rice plants grown with fertilizer treatment N0 and N6, respectively (Figure 4a), and three shading treatments (LS-0, LS-50, and LS-75) for rice plants cultivated using N12 fertilizer treatment (Figure 4b). The shading effect on the %N, %C, *δ*^13^C, *δ*^15^N, *δ*^2^H, and *δ*^18^O values of rice grown under different fertilizer treatments followed the same pattern. The light shading treatment showed a significant effect on the %N, *δ*^15^N, *δ*^13^C, and *δ*^2^H values in different rice fractions. The values of %N were significantly lower at LS-0 than that of LS-50 and LS-75 in all three rice grain fractions. In the case of PR, the highest %N value was observed LS-75 (2.3 ± 0.2%) followed by LS-50 (2.0 ± 0.2%) and the lowest was observed in LS-0 (1.6 ± 0.1%), respectively. The *δ*^15^N values followed a unique pattern. It can be seen in Figure 4b that the *δ*^15^N signature among different rice fractions was significantly higher in LS-50 compared to LS-0 (non-shaded), but it decreased in LS-75. No significant difference was found in the *δ*^15^N values for LS-0 and LS-75. Further investigation is required to explore this mechanism since no literature was found that explained this phenomenon in rice. However, previous research has shown that the effect of maize kernel shading (40%) on %N at different stages showed a decrease in the %N content (up to 60%) compared to the control (ambient sunlight), which is not consistent with our study [27]. This crop difference suggests that plant physiological characteristics can impart significant crop variation for N uptake, retention, utilization, and isotopic fractionation. 

The *δ*^13^C values among different rice fractions also showed significant variations for different shading treatments. The %C values of LS-0 in different rice fractions including PR (−28.3‰), BR (−28.8‰), and RH (−30.5‰) were significantly higher than those of LS-50 (29.5‰, −30.0‰, −31.6‰), and LS-75 (−29.7‰, −30.1‰, −31.5‰), respectively, indicating that at greater light intensity, the ^13^C isotopic fractionation is higher. In previous studies, a similar trend in *δ*^13^C values for citrus, grapefruit, and banana plants were reported under different shading treatments [10,28,29]. The *δ*^13^C values are positively correlated with irradiance and a similar trend of increasing *δ*^13^C values with increasing irradiance has been found in banana plants [28]. The *δ*^13^C values have been shown to be negatively correlated to internal CO_2_ concentration in plants during carbon uptake, so a decrease in *δ*^13^C values under shaded treatments indicates increased conductance during the time that the CO_2_ was fixed and/or a decreased photosynthetic rate [10]. Another study argued that lower *δ*^13^C values in shaded treatments clearly indicated that shading disrupted the photosynthate metabolism, reducing the photosynthate accumulation in grains [30].

The *δ*^2^H values also showed significant variations for PR and BR. In PR, the highest *δ*^2^H value (−65.3‰) was found in LS-75 followed by LS-50 (−69.3‰), and the lowest was observed in LS-0 (−77.0‰), respectively. The same trend was observed for BR and there were significant differences between LS 50/75 and LS-0. Soil water (from groundwater, irrigation water, or precipitation) is the main source of *δ*^2^H in plants, and the present results indicate that different light intensities affect the *δ*^2^H values of soil water, causing the *δ*^2^H signature of rice grains to vary through transpiration. No significant difference was observed for the *δ*^18^O values among the different rice fractions at different light intensities.

## 4. Conclusions

In this study, the effect of cultivar type, fertilizer supply, and light intensity on the distribution of elemental contents and light stable isotopes (C, N, H, and O) in different rice fractions were investigated. Significant variations were observed for *δ*^13^C, *δ*^15^N, *δ*^2^H, and *δ*^18^O values of different rice fractions among different cultivars. Fertilizer application rates showed a strong association with %N, *δ*^15^N, *δ*^2^H, and *δ*^18^O values, although there was no significant variation in the %C and *δ*^13^C values. The light intensity levels also showed a significant influence on isotopic contents among different rice fractions. The *δ*^13^C values showed a positive correlation with irradiance. The *δ*^2^H and *δ*^15^N values decreased with an increase in the irradiance. It is concluded that all factors mentioned above significantly influence isotopic values and should be considered when addressing the authenticity of rice. Furthermore, care should be taken when selecting the rice fraction for future studies as isotopic signatures vary considerably among different rice fractions. The findings from this study will have a significant impact on understanding different climatic trends (seasonal and annual) and fertilizer effects on rice and allow better understanding of isotopic variability for different geographical regions, farming practices, and environmental conditions.

## Figures and Tables

**Figure 1 foods-12-01832-f001:**
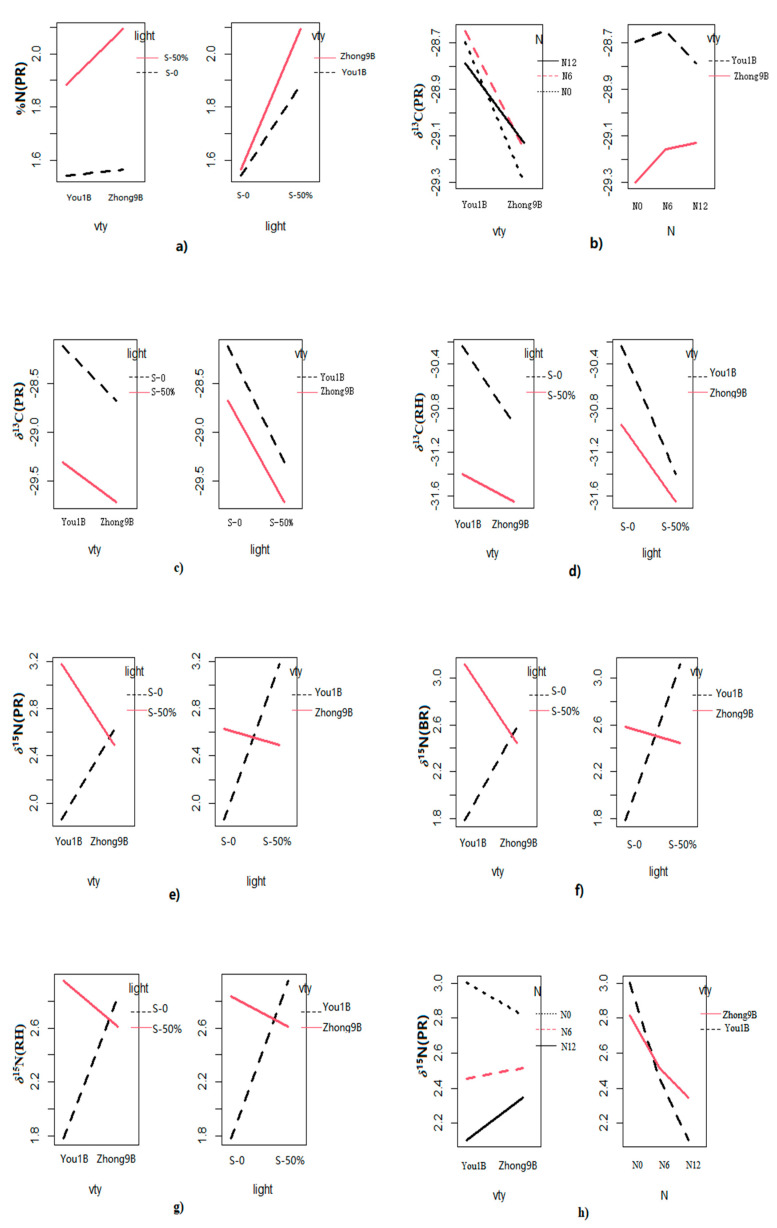
Interaction effects for the δ^13^C, δ^15^N, δ^2^H, and δ^18^O values among rice varieties (vty), fertilizer concentration (N), and light intensity of different rice fractions. (**a**) vty × light (PR, %N); (**b**) vty × N (PR, δ^13^C); (**c**) vty × light (PR, δ^13^C); (**d**) vty × light (RH, δ^13^C); (**e**) vty × light (PR, δ^15^N); (**f**) vty × light (BR, δ^15^N); (**g**) vty × light (RH, δ^15^N); (**h**) vty × N (PR, δ^15^N); (**i**) N × light (RH, δ^15^N); (**j**) vty × N (PR, δ^2^H); (**k**) vty × light (PR, δ^2^H); (**l**) vty × N (RH, δ^18^O); (**m**) vty × light (RH, δ^18^O).

**Figure 2 foods-12-01832-f002:**
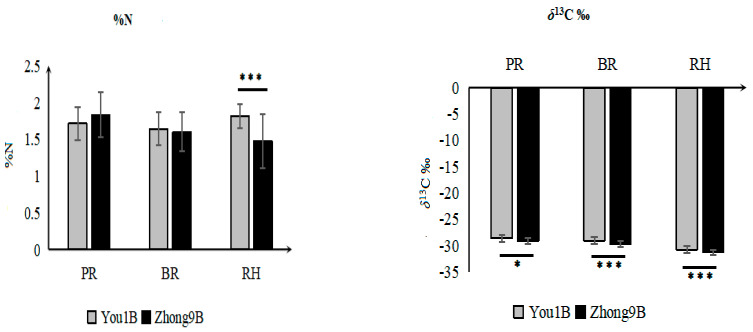
Variation of %N, *δ^1^*^3^C, *δ*^15^N, *δ*^2^H, and *δ*^18^O values in the polished rice (PR), brown rice (BR), and rice husk (RH) between two rice varieties. *** Indicates high significance at *p* < 0.001, ** *p* < 0.01, and * *p* < 0.05 (Duncan’s test).

**Figure 3 foods-12-01832-f003:**
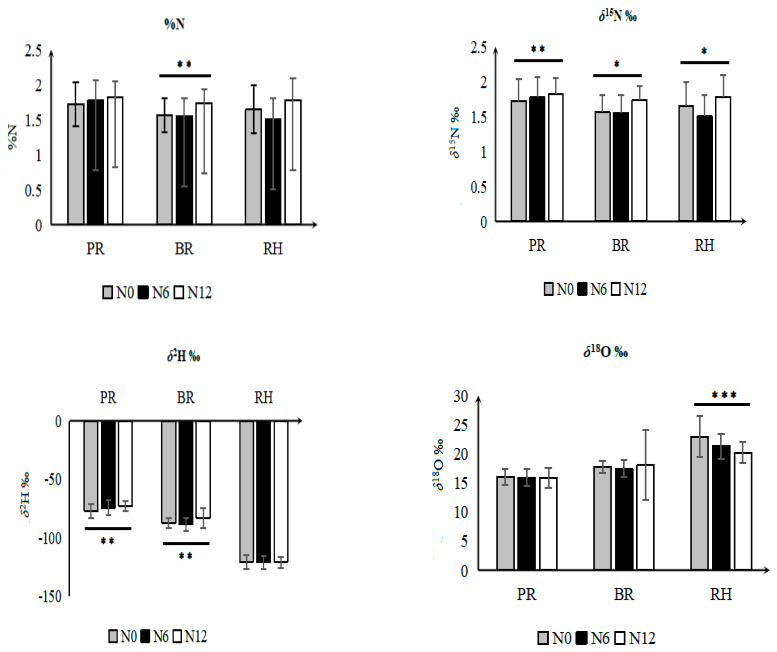
The value of %N, *δ*^15^N, *δ*^2^H, and *δ*^18^O stable isotopes in rice husk (RH), polished (PR), and brown rice (BR) under different nitrogen fertilization levels. *** Indicates high significance at *p* < 0.001, ** *p* < 0.01, and * *p* < 0.05 (Duncan’s test).

**Figure 4 foods-12-01832-f004:**
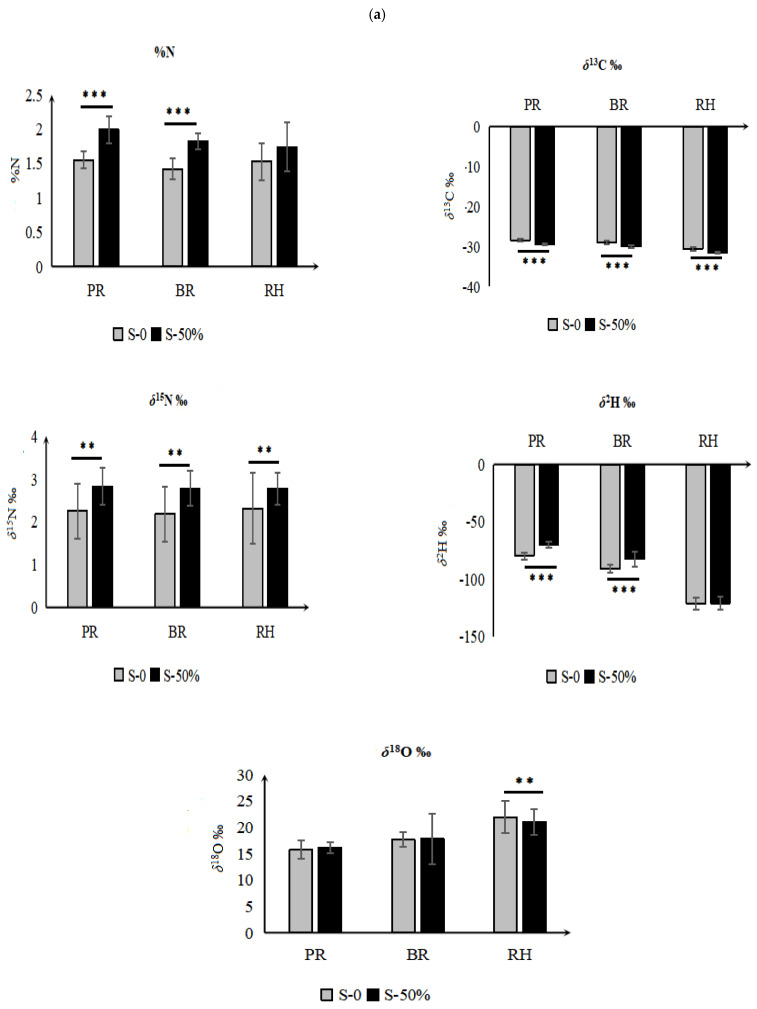
The %N, *δ*^13^C, *δ*^15^N, *δ*^2^H, and *δ*^18^O values in rice husk (RH), polished (PR), and brown rice (BR) under (**a**) two different light shading levels and (**b**) three different light shading levels. *** Indicates high significance at *p* < 0.001, ** *p* < 0.01, and * *p* < 0.05 Duncan’s test).

**Table 1 foods-12-01832-t001:** A combined analysis of variance for different influencing factors on the stable carbon (*δ*^13^C), nitrogen (*δ*^15^N), oxygen (*δ*^18^O), and hydrogen (*δ*^2^H) values among different rice fractions.

	Pr (>F)						
	Factor	%C	%N	*δ*^13^C (‰)	*δ*^15^N (‰)	*δ*^2^H (‰)	*δ*^18^O (‰)
**Polished Rice**	Fertilizer (N)	0.300	0.168	0.241	0.007 **	0.003 **	0.903
	Light shading (LS)	0.624	0.000 ***	0.000 ***	0.001 **	0.000 ***	0.408
	Variety (vty)	0.960	0.011 *	0.000 ***	0.541	0.222	0.004 **
	N × LS	0.945	0.528	0.291	0.061	0.190	0.893
	Vty × N	0.289	0.072	0.007 **	0.043 *	0.028 *	0.367
	Vty × LS	0.889	0.033 *	0.016 *	0.000 ***	0.030 *	0.456
**Brown Rice**	Fertilizer (N)	0.061	0.007 **	0.775	0.014 *	0.005 **	0.848
	Light shading (LS)	0.208	0.000 ***	0.000 ***	0.001 **	0.000 ***	0.920
	Variety (vty)	0.398	0.339	0.000 ***	0.338	0.029 *	0.683
	N × LS	0.432	0.630	0.087	0.073	0.273	0.954
	Vty × N	0.662	0.354	0.439	0.583	0.187	0.287
	Vty × LS	0.876	0.602	0.258	0.000 ***	0.114	0.403
**Rice Husk**	Fertilizer (N)	0.326	0.165	0.344	0.002 *	0.974	0.000 ***
	Light shading (LS)	0.081	0.053	0.000 ***	0.002 **	0.829	0.008 **
	Variety (vty)	0.833	0.000 ***	0.000 ***	0.004 **	0.000 ***	0.000 ***
	N × LS	0.208	0.581	0.153	0.015 *	0.541	0.283
	Vty × N	0.476	0.217	0.279	0.695	0.253	0.000 ***
	Vty × LS	0.586	0.363	0.025 *	0.000 ***	0.708	0.000 ***

Note: ******* Indicates highly significant at *p* < 0.001, ******
*p* < 0.01, and *****
*p* < 0.05.

## Data Availability

The data presented in this study are available on request from the corresponding author.

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
