# Peer review of "Effects of Light Shading, Fertilization, and Cultivar Type on the Stable Isotope Distribution of Hybrid Rice"

_foods, 2023, doi:10.3390/foods12091832_

Round 1

Reviewer 1 Report

Results are not very convincing. The statistical layout and style of the paper must be improved. Statistics must be discussed in details.

Line 74. Instead of: »effects from different", better:_ »effects of different«.

Lines 81-84. Imrove the style of this sentence. »will provide« is in this case to much promissing.

Line 93. For »hm2« use international metric units.

Subtitle 2.2.: detailed data how polished rice (PR), brown rice (BR), and rice husk 113 (RH) were obtained, are here missing.

Lines 229-238. Improve the text and compare with the relevant results of other authors in other starchy grain crops.

Line 238. Citation for this?

Author Response

Comments and Suggestions for Authors

Results are not very convincing. The statistical layout and style of the paper must be improved. Statistics must be discussed in details.

Thanks for your suggestions. We have carefully gone through the paper and improved the results and discussion section with more details and supporting evidence. We have clarified Figure captions and enhanced figures where possible. The statistics are discussed in as much detail as possible given the lack of previous research around the topics of this paper. Although, light stable isotopes are widely used for food authentication and traceability studies, there are not many research papers which investigate the many factors and mechanisms which need to be considered in order to correctly discriminate rice samples sourced from adjoining areas or where the same agricultural practices are adopted. This study was conducted to understand the effect of different climatic factors such as environmental (sunshine), botanical (cultivar type) and fertilization which can impart subtle variations on stable isotopes. Isotopic variations of different factors among different rice tissue types were also investigated as each part of the rice grain also showed unique fingerprints. This deeper understanding can now be used to develop a more robust authenticity model. One way ANOVA and Multivariate analysis (Multiway ANOVA) was used to check the significant differences among treatments and also to find the interaction effects of different factors. To the best of our knowledge no previous field studies have been conducted to explore these factors on elemental (%C, %N) and light stable isotopes (δ13C, δ15N, δ2H, and δ18O). Our results provide valuable insights regarding the analyzed parameters that should be considered when addressing the authenticity and origin of rice over single or multiple seasons.

Line 74. Instead of: »effects from different", better:_ »effects of different«.

Correction has been made. Section 1paragraph 3, line 4 page 2

Lines 81-84. Improve the style of this sentence. »will provide« is in this case to much promising.

Correction has been made. Line 82 “contribute more insight”

Line 93. For »hm2« use international metric units.

Correction has been made. All ‘hm2’ has been changed to ‘ha’

Subtitle 2.2.: detailed data how polished rice (PR), brown rice (BR), and rice husk 113 (RH) were obtained, are here missing.

Information has been added. Page 3 section 2.2 1st paragraph.

Lines 229-238. Improve the text and compare with the relevant results of other authors in other starchy grain crops.

Manuscript was edited by native speaker. Due to very limited studies it is very hard to compare results for some isotopes; however, appropriate discussion for any possible variation has been provided with mechanisim and supported with references in whole discussion. Page 7 sec. 3.2 highlighted portion, page 8 sec 3.3highlighted portion Page 9 1st, 2nd and 3rd paragraphs page 10 sec. 3.4 2nd paragraph highlighted portion.

Line 238. Citation for this?

Citation has been added. Page 9 1st paragraph last line.

Reviewer 2 Report

Authors submitted a manuscript entitled “Effects of shading, fertilization and cultivar on the stable isotope distribution of hybrid rice” to Foods journal. This paper presents a study about the effect of 3 factors (cultivar, fertilizer supply, and light intensity) on the total percentages of C and N, and light stable isotopes of the majority elements (C, N, H, and O) on rice. ANOVA was used to identify interactions among these factors. I think the study is of interest to readers. However, some revisions must be made. Although the manuscript is well written; however, the discussion needs to be deeper. The figure legends are not very well described. Overall, the study is very well balanced, and manuscripts falls to the aims and scopes of this journal. I have few comments and suggestions.

Title

Please, add the word "light" after "shading

Abstract.

line 22: please, remove "In this study".

I missed the numeric results in the abstract. Please add numeric values.

Materials and methods

L95; 50%, 30%, and 20%

L155-156; polished rice (PR), brown rice (BR), and rice husk (RH), no need to abbreviate again the same words if they are already abbreviated in previous main text. Also, L157; carbon content (% C), authors should abbreviate it when it appears first (L78), and no need to abbreviate again and again (e.g. L154). Please check it for other abbreviations also, in whole manuscript.

line 125. What was precision and error for C, N, H,and O?

Line 127. Need to list machine type, brand, etc.

Authors should provide the footer for the table 1. They should mention the meaning of two and three asterisks and values without asterisks. Authors can insert “ns” for non-significant difference. Also write down the test used, and p-value in footer.

Legend of the figure 1 is not well explained. Figure a-m should be explained. Same problem is in figure 2, Figure 3and Figure 4. Figures are not named, and legends are not well described. What is "vty"? It is variety? Please, use the same abbreviations in all the manuscript.

Results and discussion

I miss numeric values in the discussion. For example, when it is written that the % of N decreases in some conditions, how much is this decrease?

Reading the section 3.3 I got confused. The nitrogen results are consistent or not?

A deeper discussion about authenticity is necessary. Based on the results, was it necessary to use the six measured parameters? Were all of them significant?

Please compare your results to previous reports and provide proper reason to choose these parameters.

The discussion of the results is poor. Authors should speculate the results in relevance to their findings and previously reported studies. They should provide some statements about some gaps and future research directions.

Please check the whole manuscript carefully and remove the possible mistakes about abbreviations, typos, grammar and sentences.

Author Response

Comments and Suggestions for Authors

Authors submitted a manuscript entitled “Effects of shading, fertilization and cultivar on the stable isotope distribution of hybrid rice” to Foods journal. This paper presents a study about the effect of 3 factors (cultivar, fertilizer supply, and light intensity) on the total percentages of C and N, and light stable isotopes of the majority elements (C, N, H, and O) on rice. ANOVA was used to identify interactions among these factors. I think the study is of interest to readers. However, some revisions must be made. Although the manuscript is well written; however, the discussion needs to be deeper. The figure legends are not very well described. Overall, the study is very well balanced, and manuscripts falls to the aims and scopes of this journal. I have few comments and suggestions.

Title

Please, add the word "light" after "shading

Title has been revised

Abstract.

line 22: please, remove "In this study".

Change has been incorporated and highlighted. See abstract.

I missed the numeric results in the abstract. Please add numeric values.

Numeric values has been added where appropriate in abstract.

Materials and methods

L95; 50%, 30%, and 20%

Correction has been done. Page 2 sec 2.1 highlighted portion.

L155-156; polished rice (PR), brown rice (BR), and rice husk (RH), no need to abbreviate again the same words if they are already abbreviated in previous main text. Also, L157; carbon content (% C), authors should abbreviate it when it appears first (L78), and no need to abbreviate again and again (e.g. L154). Please check it for other abbreviations also, in whole manuscript.

Suggestion has been incorporated. Page 4 section 3.1 highlighted.

line 125. What was precision and error for C, N, H and O?

It has been added. Page 3 sec 2.2 highlighted portion.

Line 127. Need to list machine type, brand, etc.

It has been added. Sec 2.2 1st paragraph.

Authors should provide the footer for the table 1. They should mention the meaning of two and three asterisks and values without asterisks. Authors can insert “ns” for non-significant difference. Also write down the test used, and p-value in footer.

Suggestion has been incorporated. See table 1. Table 1 summarized the results of multi way ANOVA among different factors including interaction effect. No posthoc tests applied since MANOVA produces statistically significant results-as signified by minuscule p values. For one way ANOVA, Duncan’s test was applied to test significant differences among different treatments. See fig 2-4.

Legend of the figure 1 is not well explained. Figure a-m should be explained. Same problem is in figure 2, Figure 3and Figure 4. Figures are not named, and legends are not well described. What is "vty"? It is variety? Please, use the same abbreviations in all the manuscript.

Changes has been incorporated in figure l legends and also in the text. Page 4 sec 3.1

Results and discussion

I miss numeric values in the discussion. For example, when it is written that the % of N decreases in some conditions, how much is this decrease?

Numeric values has been added in whole discussion and highlighted as well. % decrease among nitrogen content as reported in literature has also been included

Reading the section 3.3 I got confused. The nitrogen results are consistent or not?

Results were consistent with previous studies and possible variation mechanism is discussed. 2nd and 3rd paragraph, page 9.

A deeper discussion about authenticity is necessary. Based on the results, was it necessary to use the six measured parameters? Were all of them significant?

All factors including cultivar type, fertilization and light intensity level showed significant influence on analyzed parameters (%C, %N, δ13C, δ15N, δ2H, and δ18O). In addition, appropriate discussion for any possible variation has been provided with mechanism and supported with references in whole discussion.  Page 7 sec. 3.2 highlighted portion, page 8 sec 3.3highlighted portion Page 9 1st, 2nd and 3rd paragraphs, page 10 sec. 3.4 2nd paragraph highlighted portion.

Please compare your results to previous reports and provide proper reason to choose these parameters.

In recent years, elemental and light stable isotopes have gained increasing interest in food authentication studies; however, there are very few studies that have focused on the different factors which could impart variation for these isotopes. Before a robust model can be made, it is necessary to understand how stable isotopes vary with these variations (environmental, genetic, and agronomic practices) to better predict rice origin and authenticity using our models. Section 1 paragraph 2 & 3.   The results were compared with available literature and  discussion has been improved.

The discussion of the results is poor. Authors should speculate the results in relevance to their findings and previously reported studies. They should provide some statements about some gaps and future research directions.

 Due to very limited studies it is very hard to compare results for some isotopes; however, most of the results were compared with available literature See section 3.2, 3.3 and 3.4 yellow heighted and appropriate discussion for any possible variation has been provided with mechanism and supported with references in whole discussion and highlighted in red.

See last sentence in Conclusions

Please check the whole manuscript carefully and remove the possible mistakes about abbreviations, typos, grammar and sentences.

Manuscript is edited by native speaker.

Reviewer 3 Report

Very good paper, well written with interesting data. My main point is, that total N content presented by other authors in husk is much lower, e.g. Pure Appl. Bio., 1(1): 14-15, June- 2012  or Semie Kim, Young-Il Lim, in Computer Aided Chemical Engineering, 2022. Please clarify.

Also accuracy of measurement of 1.0 mg of powdered sample is problematic. And last issue, that should be clarified, please provide  the moisture ratio and water content of investigated  samples. Changes or just water content cold impact on final δ 2H and δ 18O values.

Author Response

Comments and Suggestions for Authors

Very good paper, well written with interesting data. My main point is, that total N content presented by other authors in husk is much lower, e.g. Pure Appl. Bio., 1(1): 14-15, June- 2012  or Semie Kim, Young-Il Lim, in Computer Aided Chemical Engineering, 2022. Please clarify.

Rice from China has historically known for higher %N content as it is bred to have higher protein content. This statement is taken from KORENAGA et al. [Statistical Analysis of Rice Samples for Compositions of Multiple Light Elements (H, C, N, and O) and Their Stable Isotopes, ANALYTICAL SCIENCES AUGUST 2010, VOL. 26, pg 873] The nitrogen content of most rice samples centered at 1.0%, with some exceptions. For instance, USA's long grain rice (2005LF) and Chinese rice were as high as 1.3%.

Also accuracy of measurement of 1.0 mg of powdered sample is problematic.

For the N content and isotope analysis, around 4.5 to 5.5 mg of sample was used, not 1.0 mg of sample

And last issue, that should be clarified, please provide  the moisture ratio and water content of investigated  samples. Changes or just water content cold impact on final δ 2H and δ 18O values.

The samples were dried under vacuum freeze-drying for three days at 60 C to remove all exchangeable water, so that the water content did not impact the final δ2H and δ18O values